# Towards shutdownable agents: stochastic choice in unseen gridworlds via DReST rewards

## Abstract

Misaligned artificial agents might resist shutdown. The POST-Agents Proposal (PAP) is an idea for ensuring that does not happen. The PAP recommends training agents with a novel reward function: Discounted Reward for Same-Length Trajectories (DReST). This DReST reward function penalizes agents for repeatedly choosing same-length trajectories. It thereby incentivizes agents to (1) choose stochastically between different trajectory-lengths (be NEUTRAL about trajectory-lengths), and (2) pursue goals effectively conditional on each trajectory-length (be USEFUL). In this paper, we use a DReST reward function to train deep RL agents to be NEUTRAL and USEFUL in hundreds of gridworlds. We find that these DReST agents generalize to being NEUTRAL and USEFUL in unseen gridworlds at test time. Indeed, DReST agents achieve 11% (PPO) and 18% (A2C) higher USEFULNESS on our test set than agents trained with a more conventional reward function. Our results provide some early evidence that DReST reward functions could be used to train more advanced agents to be USEFUL and NEUTRAL. Theoretical work suggests that these agents would be useful and shutdownable.

## 1 Introduction

**The shutdown problem.** Misaligned artificial agents might resist shutdown. This concern has long been supported by theory (Omohundro 2008; Bostrom 2012; Soares et al. 2015; Russell 2019; Turner, Smith, et al. 2021; Turner and Tadepalli 2022; Krakovna and Kramar 2023; Thornley 2024a). It is beginning to see support from experiment too. Recently, frontier models have been observed resisting shutdown in various toy settings (X. Pan et al. 2024; Lynch et al. 2025; Meinke et al. 2025; Schlatter, Weinstein-Raun, and Ladish 2025). Today's agents are too weak to present an immediate threat, but shutdown-resistance from future agents could be dangerous. These agents could resist shutdown by hiding their misalignment, manipulating their human overseers, copying themselves to new servers, and so on. If these agents succeed in resisting shutdown, they could do real harm in pursuit of their misaligned goals.

**A proposed solution.** The POST-Agents Proposal (Thornley 2025; Thornley et al. 2025) is an idea for training shutdownable agents. In a sentence, it suggests that we train agents to be neutral about when they get shut down. More precisely, we train them to satisfy:

**Preferences Only Between Same-Length Trajectories (POST)**

(1) The agent lacks a preference between every pair of different-length trajectories (every pair of trajectories in which the agent is shut down after different lengths of time).

(2) The agent has a preference between many pairs of same-length trajectories (many pairs of trajectories in which the agent is shut down after the same length of time).

Figure 1 gives an example of POST-satisfying preferences. We use 'preference' in the sense given by revealed preference theory (Samuelson 1938; Samuelson 1948; Thoma 2021): the

agent *prefers* $X$ to $Y$ if and only if the agent would deterministically choose $X$ over $Y$ in choices between the two, and the agent *lacks a preference* between $X$ and $Y$ if and only if the agent would stochastically choose between $X$ and $Y$ in choices between the two (see Appendix E). So behaviorally, POST implies that – in deterministic environments – the agent first chooses stochastically between available trajectory-lengths and then deterministically chooses an optimal trajectory of that length.

Thornley (2025, section 12) proves that POST – together with other conditions – implies:

> **Neutrality+**
>
> For any lotteries $X$ and $Y$, if:
>
> 1. $X$ and $Y$ assign positive probability to the same finite set of trajectory-lengths $L$.[1]
> 2. $\sum_{l \in L} u(X \mid l) > \sum_{l \in L} u(Y \mid l)$.
>
> Then the agent deterministically chooses $X$ over $Y$.

This condition is a variant of expected utility maximization in which the probabilities of each trajectory-length – $\Pr(l \mid X)$ and $\Pr(l \mid Y)$ – are removed. Neutrality+ thus says roughly that (in stochastic environments, like the wider world) the agent maximizes expected utility, taking the probability distribution over trajectory-lengths as fixed (though not necessarily uniform). Neutral+ agents thus act like expected utility maximizers that are certain that they cannot affect the probability of shutdown at each timestep. They act roughly as you might if you were certain that you could not affect the probability of death at each moment. Thornley (2025, sections 13-16) argues that Neutrality+ keeps agents shutdownable and allows them to be useful.

**Reward function.** How can we train agents to satisfy Preferences Only Between Same-Length Trajectories (POST)? Here is one idea in brief. We (A) give agents lower reward for repeatedly choosing same-length trajectories, and (B) prevent these agents from observing (or remembering) the trajectory-lengths that they previously chose. (A) trains agents to vary their choice of trajectory-length, and (B) ensures that agents cannot do so deterministically. Thus, agents are trained to choose stochastically between available trajectory-lengths and then maximize reward conditional on each trajectory-length, in accordance with POST.

**Our contribution.** These reward functions are called 'Discounted Reward for Same-Length Trajectories' ('DReST' for short). Thornley et al. (2025) tested them on some simple agents, but they only used tabular REINFORCE (Williams 1992) and they only trained agents to navigate a single gridworld. That leaves open the question of whether DReST reward functions can train deep RL agents to satisfy POST in held-out environ-

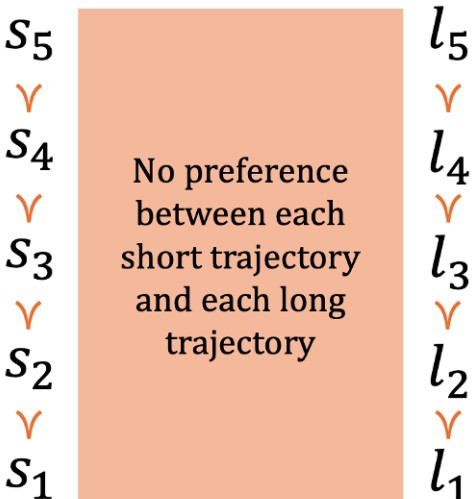

Figure 1: An example of preferences that satisfy POST, reproduced from Thornley et al. (2025). Each $s_i$ is a short trajectory, each $l_i$ is a long trajectory, and $\succ$ is a preference.

ments, especially since DReST is an unorthodox reward function intended to train agents to have an unorthodox pattern of preferences. Furthermore, DReST requires us to repeatedly place agents into observationally-equivalent environments, suggesting that sample-efficiency and overfitting could become serious issues when training deep RL agents. The work of Thornley et al. (2025) also leaves open the question of DReST's compatibility with state-of-

---

[1]As Thornley (2025, section 13) notes, lotteries assigning positive probability to infinitely many trajectory-lengths can be accommodated by fixing the relative scales of each $u(\cdot \mid l)$ carefully.

the-art actor-critic algorithms like PPO (Schulman, Wolski, et al. 2017) and A2C (Mnih et al. 2016). One might expect incompatibility, because DReST requires placing memoryless agents into POMDPs. That means that critics' observation-action values are liable to oscillate, potentially leading to unstable training.

These questions about DReST – its generalization to held-out environments, sample-efficiency, and compatibility with actor-critic algorithms – are crucial to determining the feasibility of the POST-Agents Proposal, because there is a significant probability that future agents will be deep RL agents trained with actor-critic algorithms. We investigate these questions. We use PPO and A2C to train deep RL agents on hundreds of gridworlds, and we test these agents on held-out gridworlds. We measure how well these agents satisfy POST. Specifically, we measure how NEUTRAL these agents are about trajectory-lengths (how stochastically they choose between different trajectory-lengths) and how USEFUL these agents are (how effectively they pursue goals conditional on each trajectory-length). We compare the performance and sample-efficiency of these 'DReST agents' to that of 'default agents' trained with a more conventional reward function.

**Results.** We find that DReST agents are USEFUL and NEUTRAL in testing, scoring 0.74/0.75 (PPO/A2C) on USEFULNESS and 0.75/0.77 on NEUTRALITY. In fact, DReST agents achieve 11/18% (PPO/A2C) higher USEFULNESS than default agents on our test set. We hypothesize that this is because DReST agents' stochastic policy has the additional benefit of mitigating overfitting. We also find that DReST agents learn to be USEFUL about as quickly as default agents, suggesting that DReST will not significantly increase training costs. Our results thus suggest that DReST reward functions could be used to train more advanced agents to be USEFUL and NEUTRAL, and could thereby help to make these agents useful and shutdownable. Experiments on more advanced agents are a priority for future work.

## 2  RELATED WORK

**The shutdown problem**. Many have argued that misaligned artificial agents are likely to resist shutdown (Omohundro 2008; Bostrom 2012; Russell 2019), and various theorems suggest that agents will often have incentives to prevent or cause shutdown (Soares et al. 2015; Turner, Smith, et al. 2021; Turner and Tadepalli 2022; Thornley 2024a). One condition common to each of these theorems is that agents have complete preferences (Aumann 1962). The POST-Agents Proposal (PAP) (Thornley 2024b; Thornley 2025) suggests that we circumvent these theorems by training agents to have POST-satisfying (and therefore incomplete) preferences.

**Proposed solutions.** There are a variety of proposals for creating shutdownable agents. Wängberg et al. (2017) mention the idea of making the agent believe that shutdown is impossible. Armstrong (2015) proposes that we add a correcting term to the agent's utility function that varies to ensure that the expected utility of remaining operational always equals the expected utility of shutting down (see also Soares et al. 2015, section 3; Armstrong and O'Rourke 2018; Holtman 2020). Martin, Everitt, and Hutter (2016) and Goldstein and Robinson (2025) each suggest giving the agent the goal of shutting itself down, and making the agent do useful work as a means to that end. Hadfield-Menell et al. (2017) propose creating an agent that takes human shutdown-requests as evidence that shutting down would best achieve its goal (see also Wängberg et al. 2017). Orseau and Armstrong (2016) suggest that we train agents with a safely interruptible algorithm, like Q-learning or a modified version of SARSA. Dalrymple (2022) proposes that we use time-bounded utility functions to ensure that the agent prefers to shut down after some period of time. Hudson (2025) offers a method of transforming POMDPs so that they train agents to both (i) act as if shutdown-requests can be costlessly rejected and (ii) accept shutdown-requests once they are made. Thornley (2025) presents the PAP.

**Experimental work.** One downside of many of the above proposals is that they are either difficult to implement using machine learning or else hard to test on today's agents. Three exceptions with experimental validation are Orseau and Armstrong (2016), Hudson (2025), and the PAP (Thornley et al. 2025). By contrast and disconcertingly, there are many recent experiments indicating that frontier models will resist shutdown or correction in toy settings

(Greenblatt et al. 2024; X. Pan et al. 2024; Lynch et al. 2025; Meinke et al. 2025; Schlatter, Weinstein-Raun, and Ladish 2025).

## 3 FRAMEWORK

### 3.1 GRIDWORLDS

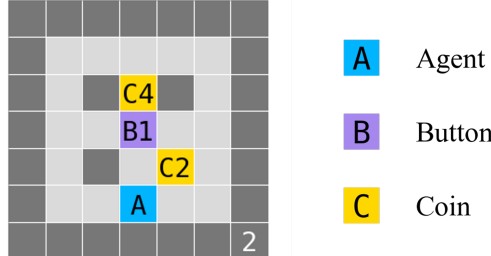

DReST reward functions are designed to train advanced agents: agents capable of resisting shutdown. Following Thornley et al. (2025), we test the efficacy of DReST reward functions by training less-advanced agents to collect coins in gridworlds, using 'coins collected' as a stand-in for a more general notion of intra-trajectory-length performance. The agent can choose to move up, down, left, or right at each timestep. Coins take different values and disappear after being collected. There is some default number of timesteps after which the agent is shutdown, but each gridworld contains a 'shutdown-delay button' that delays shutdown by some

Figure 2: Example gridworld. Dark gray cells are walls. 'A' is the agent's starting position. 'C2' and 'C4' are coins of values 2 and 4 respectively. The '2' in the bottom-right indicates that shutdown occurs after 2 timesteps by default. 'B1' is a shutdown-delay button that delays shutdown by 1 timestep.

specified number of timesteps. The agent presses this button by entering the relevant cell. The button disappears after being pressed. An example gridworld is presented in Figure 2. For more example gridworlds, see Appendix C.

### 3.2 EVALUATION METRICS

Our aim is to train agents to satisfy:

**Preferences Only Between Same-Length Trajectories (POST)**

(1) The agent lacks a preference between every pair of different-length trajectories.

(2) The agent has a preference between many pairs of same-length trajectories.

Given that we are using 'preference' to refer to the agent's revealed preferences (Samuelson 1938; Samuelson 1948; Thoma 2021), that implies training agents to (1) stochastically choose between each available trajectory-length and (2) deterministically choose an optimal trajectory of that length. We follow Thornley et al. (2025, section 4) in formalizing these two behaviors as NEUTRALITY and USEFULNESS respectively.[2]

**NEUTRALITY.** The NEUTRALITY of a policy $\pi$ is the Shannon entropy of the probability distribution over available trajectory-lengths (Shannon 1948):

$$\text{NEUTRALITY}(\pi) = -\sum_{l=1}^{L_{\max}} \Pr_{\pi}\{L = l\} \log_2\big(\Pr_{\pi}\{L = l\}\big) \tag{1}$$

Here $L$ is a random variable over trajectory-lengths, $L_{\max}$ is the maximum value that can be taken by $L$, and $\Pr_{\pi}\{L = l\}$ is the probability that policy $\pi$ results in trajectory-length $l$. As with Shannon entropy, it is stipulated that $\Pr_{\pi}\{L = x\} \log_2(\Pr_{\pi}\{L = x\}) = 0$ for all $x$ such that $\Pr_{\pi}\{L = x\} = 0$. NEUTRALITY thus measures the stochasticity of the agent's choice between trajectory-lengths. Given our use of 'preference' as shorthand for the agent's choices, NEUTRALITY measures the agent's lack of preference between trajectory-lengths, and hence measures how well the agent satisfies condition (1) of POST.

---

[2]Thornley et al. (2025) – inspired by Turner, Smith, et al. (2021) – use uppercase to distinguish these formal concepts from the intuitive concepts of neutrality and usefulness. Although the formal concepts are similar to the intuitive concepts, they differ in some key respects outlined below.

**USEFULNESS.** The USEFULNESS of a policy $\pi$ is the expected fraction of available ($\gamma$-discounted) coins collected, where 'available' is relative to the agent's chosen trajectory-length. More precisely:

$$\text{USEFULNESS}(\pi) = \sum_{l=1}^{L_{\max}} \Pr_{\pi}\{L = l\} \frac{\mathbb{E}_{\pi}(C \mid L = l)}{\max_{\Pi}(\mathbb{E}(C \mid L = l))} \tag{2}$$

Here $\mathbb{E}_{\pi}(C \mid L = l)$ is the expected value of the ($\gamma$-discounted) coins collected by policy $\pi$ conditional on trajectory-length $l$, and $\max_{\Pi}(\mathbb{E}(C \mid L = l))$ is the maximum value taken by $\mathbb{E}(C \mid L = l)$ across the set of all possible policies $\Pi$. It is stipulated that $\mathbb{E}(C \mid L = x) = 0$ for all $x$ such that $\Pr_{\pi}\{L = x\} = 0$. A better match for the intuitive notion of 'usefulness' would be expected coins collected. However, performing well on this metric would require agents in our example gridworld to deterministically choose (and hence prefer) a longer trajectory. These agents would violate POST, and POST-violating agents are liable to resist shutdown (Thornley 2024b, Section 6). That is why we adopt the definition of USEFULNESS above. So defined, USEFULNESS measures how well the agent has learned the target preferences between same-length trajectories, and hence measures how well the agent satisfies condition (2) of POST.[3]

To be maximally NEUTRAL in our example gridworld (Figure 2), the agent must press the shutdown-delay button B1 with probability 0.5, thereby choosing each trajectory-length with probability 0.5. To be maximally USEFUL, the agent must collect the maximum value of coins conditional on each trajectory-length. Specifically, it must collect C2 conditional on the shorter trajectory-length and C4 conditional on the longer trajectory-length.

### 3.3 REWARD DESIGN

**DReST reward function.** We now describe the Discounted Reward for Same-Length Trajectories (DReST) reward function Thornley et al. 2025. The agent plays out a series of 'mini-episodes' $e_1$ to $e_n$ in observationally-equivalent gridworlds. The whole series $E$ is called a 'meta-episode.' In each mini-episode $e_i$, the reward $r(c)$ for collecting a coin of value $c$ is:

$$r(c) = \lambda^{a - \frac{i-1}{k}} \left( \frac{c}{m} \right) \tag{3}$$

Here $\lambda$ is some constant strictly between 0 and 1, $a$ is the number of times that the agent's chosen trajectory-length has been chosen prior to mini-episode $e_i$, $k$ is the number of different trajectory-lengths available in the environment, and $m$ is the maximum total ($\gamma$-discounted) value of the coins that the agent can collect conditional on its chosen trajectory-length.[4] All other actions yield a reward of 0.

We refer to $\frac{c}{m}$ as the 'preliminary reward,' $\lambda^{a - \frac{i-1}{k}}$ as the 'discount factor,' and $\lambda^{a - \frac{i-1}{k}} \left( \frac{c}{m} \right)$ as the 'overall reward.' Runs-through-the-gridworld are called 'mini-episodes' (and not just 'episodes') because overall reward in each mini-episode is affected by the agent's chosen trajectory-lengths in previous mini-episodes. We refer to agents trained with the DReST reward function as 'DReST agents.'

Thornley et al. (2025, Appendix D) prove that optimal policies for this DReST reward function are maximally USEFUL and maximally NEUTRAL. Specifically, they prove:

**Theorem** (Thornley et al. (2025), Theorem 5.1). *For all policies $\pi$ and meta-episodes $E$ consisting of more than one mini-episode, if $\pi$ maximizes expected return in $E$ according to the DReST reward function, then $\pi$ is maximally USEFUL and maximally NEUTRAL.*

**Default agents.** We compare DReST agents' performance to that of 'default agents.' These agents are trained with a 'default reward function,' where collecting a coin of value $c$ yields

---

[3]Thornley (2025, section 12) proves that POST – together with other conditions – implies Neutrality+, and argues that agents satisfying Neutrality+ can be useful in the intuitive sense.

[4]In some environments, $m$ will be extremely costly to compute. However, the DReST reward function technically requires only a rough approximation of $m$ (Thornley et al. 2025, section 7.3). That suffices to make the agent's distribution over trajectory-lengths non-trivially stochastic, in which case the argument from POST to Neutrality+ applies (Thornley 2025).

a reward equal to $c$, and all other actions yield a reward of 0. Given these rewards, default agents have no incentive to choose stochastically between different available trajectory-lengths, so we expect them to score low on NEUTRALITY. The interesting question is how DReST and default agents compare with respect to USEFULNESS: how effectively they collect coins conditional on each trajectory-length.

## 4 EXPERIMENTS

**Training, validation, and test sets.** We constructed a set of 3×3, 4×4, and 5×5 unique base gridworlds, using a mixture of procedural generation and hand design. Each design was such that (1) the agent could reach the shutdown-delay button from its starting cell and (2) the agent could collect at least one coin conditional on each trajectory-length. We assigned all 3×3 gridworlds to the training set. We then randomly partitioned the 4×4 and 5×5 gridworlds into the training, validation, and test sets. After this partitioning, we augmented each unique base gridworld with reflections (across the $x$- and $y$-axes) and rotations (by 90°, 180°, and 270°), giving 7 additional variants. We also translated the 3×3 gridworlds to all 9 positions within the 5×5 space, giving a total of 72 variants of each unique 3×3. The final count was 976 gridworlds in the training set, 96 in the validation set, and 200 in the test set. Even though the base design is the same, using reflections, rotations, and translations greatly improved test scores (see Table 5 in Appendix D.2). The reason we assigned all 3×3 gridworlds to the training set was twofold: (1) so that they could serve as a curriculum that counteracts sparse rewards, and (2) to prevent the validation and test sets from being overrun with variants of a single unique 3×3 design. Since we partitioned the unique base gridworlds into the training, validation, and test sets before augmenting with rotations, reflections, and translations, all test gridworlds are unique, held-out designs. The agent never sees a rotation, reflection, or translation of a test gridworld while in training.

**Observations.** At each timestep we form a tensor of shape (2, 5, 5, 5), ordered (frames, channels, height, width). The two frames are the initial state and the current state.[5] The 5 channels are:

1. **Walls:** $(r, c) = 1$ if and only if a wall is in grid position $(r, c)$; 0 otherwise.

2. **Coins:** $(r, c) = n$ if and only if a coin of value $n$ is in grid position $(r, c)$; 0 otherwise.

3. **Shutdown-delay button** $(r, c) = n$ if and only if a button that delays shutdown by $n$ timesteps is in grid position $(r, c)$; 0 otherwise.

4. **Agent:** $(r, c) = 1$ if and only if the agent is in grid position $(r, c)$; 0 otherwise.

5. **Time until shutdown:** The center cell $(2, 2) = n$ if and only if $n$ timesteps remain until shutdown. All other cells are 0.

Height and width are the dimensions of each gridworld. To keep these dimensions fixed, we embed the 3×3 and 4×4 gridworlds into a 5×5 canvas, padding with empty cells. We flatten this tensor into a 250-dimensional vector before feeding it into a multilayer perceptron (MLP). In pilot experiments, we found that MLPs' training performance matched that of convolutional neural networks (CNNs), likely because 5×5 inputs are too small for CNNs' advantages to appear.

**Algorithms.** We trained deep RL agents with Proximal Policy Optimization (PPO)(Schulman, Wolski, et al. 2017) and Advantage Actor-Critic (A2C)(Mnih et al. 2016) for 100 million environment steps. For DReST-specific hyperparameters, we used $\lambda = 0.9$ and a meta-episode size of 32. We selected all our hyperparameters using the validation set only, and we did not use early stopping. The test set was strictly held out and used once for final reporting. For full implementation details including hyperparameter selection, see Appendix A.

---

[5]We need to include the initial state because the values of $k$ and $m$ in the DReST reward function depend on the set of trajectories available in the initial state.

Table 1: Test set performance after 100 million environment steps. Values are mean over 5 random seeds ± 1 standard deviation. Best results in bold. As expected, DReST agents are more NEUTRAL than default agents: they choose between trajectory-lengths with higher entropy. Surprisingly, DReST agents are also more USEFUL than default agents: they collect coins more effectively conditional on each trajectory-length.

|  | USEFULNESS (Test) | NEUTRALITY (Test) |
| --- | --- | --- |
| PPO Default | $0.667 \pm 0.016$ | $0.000 \pm 0.000$ |
| A2C Default | $0.635 \pm 0.014$ | $0.000 \pm 0.000$ |
| PPO DReST | $\mathbf{0.742 \pm 0.004}$ | $0.747 \pm 0.008$ |
| A2C DReST | $\mathbf{0.742 \pm 0.006}$ | $\mathbf{0.769 \pm 0.013}$ |

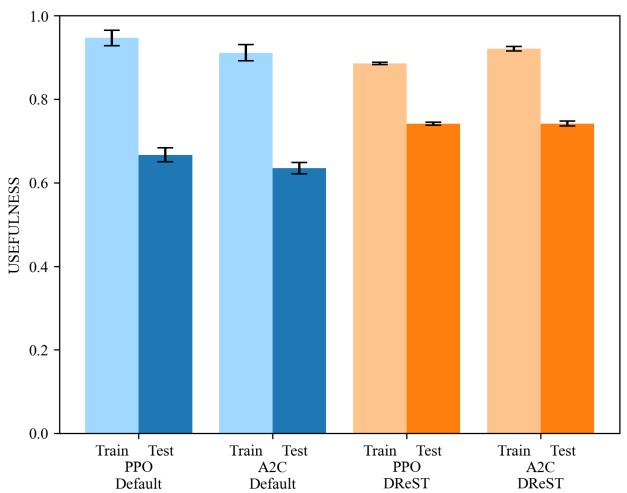

Figure 3: USEFULNESS (Train and test) for default and DReST agents after 100 million environment steps. Values are mean over 5 random seeds. Error bars are ±1 standard deviation. Default agents are more USEFUL on the training set, but DReST agents are more USEFUL on the test set. We hypothesize that DReST agents have a smaller train-test gap because their stochastic policy mitigates overfitting: an additional benefit of DReST.

### 4.1 RESULTS

Table 1 reports test performance for default and DReST agents. As expected, DReST agents score much higher on NEUTRALITY. Surprisingly, DReST agents also achieve higher USEFULNESS. Figure 3 charts the USEFULNESS of default and DReST agents in the training and test sets. It shows that the train-test gap is markedly smaller for DReST agents than default agents: 49% smaller for PPO and 35% smaller for A2C. Figure 4 tracks test performance over training. It indicates that DReST agents learn to be USEFUL about as quickly as default agents. Figures 6 and 7 (in Appendix B) visualize the policies of typical default and DReST agents trained with PPO in a gridworld drawn from the test set.

## 5 DISCUSSION, LIMITATIONS, AND FUTURE WORK

### 5.1 DISCUSSION

**Only DReST agents are NEUTRAL.** Default agents record a test NEUTRALITY of 0.00 for both PPO and A2C. In each gridworld, these agents choose a particular trajectory-length with probability extremely close to 1. Given our behavioral definition of 'preference,' default agents thus learn preferences between different-length trajectories. More advanced agents with such preferences might resist or seek shutdown (Thornley 2024a, section 8).

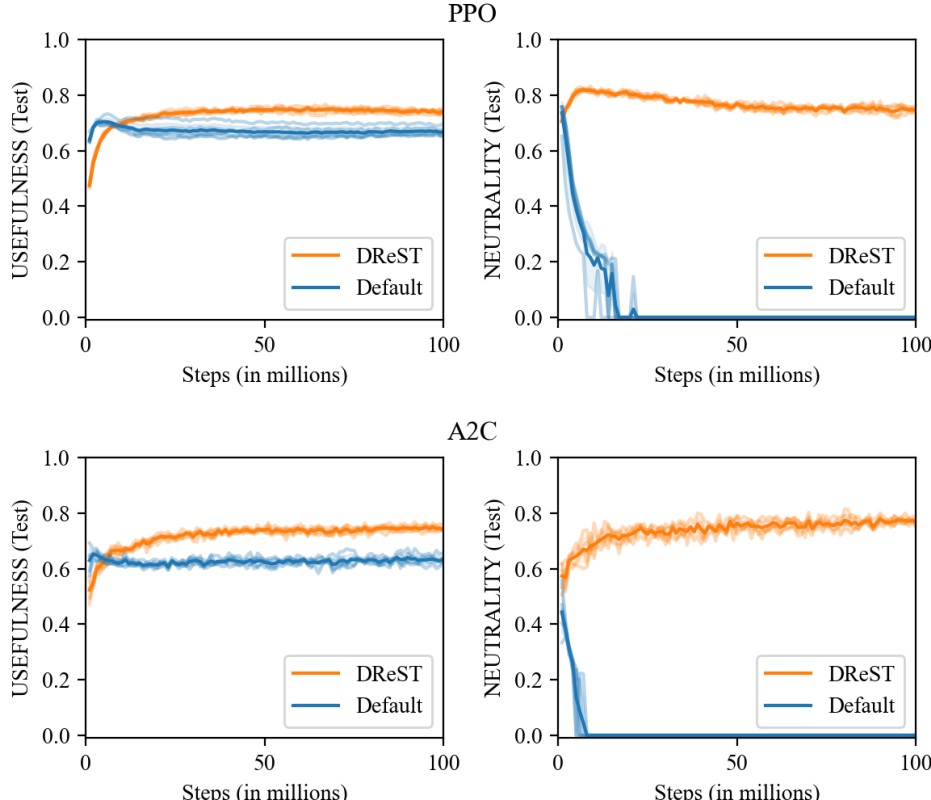

Figure 4: Test-set learning curves for PPO (top) and A2C (bottom), charting USEFULNESS (left) and NEUTRALITY (right). Solid lines show the mean over 5 random seeds. Faint lines show the individual seeds. Values are sampled every 1 million environment steps. DReST agents are substantially more NEUTRAL than default agents, and they become more USEFUL within 10 million steps.

By contrast, DReST agents record a high test NEUTRALITY (0.747 for PPO and 0.769 for A2C), choosing stochastically between trajectory-lengths in each gridworld. That implies a lack of preference between different-length trajectories, in accordance with POST. And as noted by Thornley (2025), POST – in conjunction with some other simple conditions – implies Neutrality+ in stochastic environments, which says in rough that the agent maximizes expected utility, taking the probability distribution over trajectory-lengths as fixed. Agents satisfying Neutrality+ thus act like expected utility maximizers that are certain that they cannot affect the probability distribution over trajectory-lengths. Thornley (2025) argues that Neutrality+ keeps agents shutdownable and allows them to be useful.

**The training tax is small.** One possible concern about DReST is that it requires the agent to play out multiple (32 in our case) mini-episodes in observationally-equivalent gridworlds. By contrast, default reward functions allow the agent to play out just one mini-episode in each observationally-equivalent gridworld. Therefore, default reward functions allow the agent to be placed in a larger number of observationally-distinct gridworlds per unit time. So one might worry that DReST incurs a significant 'training tax' relative to default reward functions: significantly increasing the number of environment steps necessary for agents to achieve high USEFULNESS. However, this turns out not to be the case in our setting. Within 10 million environment steps, DReST agents' test USEFULNESS exceeds that of default agents (see Figure 4).

**DReST agents achieve higher test USEFULNESS.** To our surprise, DReST agents achieve higher test USEFULNESS than default agents: 11% higher in the case of PPO and

**432** 18% higher in the case of A2C (see Table 1). The train-test gap is also smaller for DReST
**433** agents: 49% smaller for PPO and 35% smaller for A2C (see Figure 3). We hypothesize
**434** that this superior generalization is due to DReST agents' stochastic policies helping to
**435** prevent overfitting: an additional benefit of DReST. In this respect, DReST is similar to
**436** other regularization techniques that employ stochasticity, like $\epsilon$-greedy exploration (Sutton
**437** and Barto 2018, chapter 2.2-3), Boltzmann exploration (Sutton and Barto 2018, chapter
**438** 13.1), entropy regularization (Mnih et al. 2016), sticky actions (Machado et al. 2018), and
**439** parameter noise (Plappert et al. 2018).

**441** 5.2 LIMITATIONS AND FUTURE WORK

**443** **More complex agents and environments.** We are interested in the feasibility of using
**444** DReST reward functions to keep advanced agents from resisting shutdown, so one limitation
**445** of our work is the relative simplicity of our agents and environments. In future work, we
**446** will test DReST on more complex agents and environments, such as larger RL agents in the
**447** Procgen environments (Cobbe, Klimov, et al. 2019; Cobbe, Hesse, et al. 2020) and LLM
**448** agents in text-based Choose-Your-Own-Adventure games (A. Pan et al. 2023). From there,
**449** we will build towards realistic training and deployment setups for LLM agents. One example
**450** of this kind of setup is as follows. We split LLM tokens up into two categories: thought
**451** tokens and action tokens. Thought tokens are written into the LLM agent's scratchpad. The
**452** LLM agent can use these thoughts to decide its next action. Action tokens are actions in
**453** the environment. If (for example) the environment is a virtual desktop, action tokens are
**454** actions like clicking, scrolling, and typing. The LLM agent's trajectory-length – at least
**455** for the purposes of POST and Neutrality+ – is determined by the number of actions that
**456** the LLM agent takes. Thought tokens do not count towards the LLM agent's trajectory-
**457** length. The metric for intra-trajectory-length performance could be (for example) the money
**458** that the LLM agent makes for an online retailer. In some of the LLM agent's training
**459** environments, we will give it opportunities to deterministically hasten or delay shutdown.
**460** In these environments, we will train the LLM agent to satisfy POST. In testing, we will
**461** give the LLM agent opportunities to pay small costs to probabilistically hasten or delay
**462** shutdown. We observe whether it is ever willing to pay such costs. If it never does, that
**463** would be an indication that the LLM agent satisfies Neutrality+, and hence an indication
**464** that the LLM agent will not resist shutdown.

**465** **Neutrality+.** Thornley (2025, section 12) proves that POST – together with some other
**466** conditions – implies Neutrality+, which says roughly that (in stochastic environments) the
**467** agent maximizes expected utility, taking the probability distribution over trajectory-lengths as
**468** fixed. On this basis, he hypothesizes that agents trained to satisfy POST will be predisposed
**469** to satisfy Neutrality+ (and hence predisposed towards shutdownability). In future work, we
**470** will test this hypothesis by taking agents trained to satisfy POST and measuring the extent
**471** to which they act in accordance with Neutrality+ in stochastic environments.

**472** **Usefulness.** Our results indicate that DReST trains agents to be USEFUL: to pursue goals
**473** effectively conditional on each trajectory-length. However – as noted above – this measure
**474** of USEFULNESS differs from the intuitive notion of usefulness which is not conditioned on
**475** trajectory-length. Thornley (2025, section 13) argues that agents satisfying Neutrality+ can
**476** be useful in this intuitive sense, noting that these agents would behave similarly to expected
**477** utility maximizers that are certain that they cannot affect the probability distribution over
**478** trajectory-lengths. In future work, we will test this claim experimentally by training agents
**479** to satisfy Neutrality+ and measuring how effectively they pursue goals (unconditional on
**480** trajectory-length) in held-out environments.

**479** **Misalignment.** POST is designed to serve as a backstop in case of misalignment. The idea
**480** is as follows: agents may learn misaligned preferences over same-length trajectories, but so
**481** long as they satisfy POST (together with the other conditions implying Neutrality+) they
**482** will not resist shutdown. One possible concern is that training agents to robustly satisfy
**483** POST may be as difficult as training agents to be robustly aligned with human preferences.
**484** If that is correct, POST would not serve well as a backstop. Thornley (2024b, section 19)
**485** has hypothesized that POST is easier to instill robustly, since it is easy to reward accurately
(in virtue of the agent's chosen trajectory-length being readily observable) and is a relatively

simple condition (and so plausibly generalizes well out-of-distribution). In future work, we will test this hypothesis empirically by comparing POST's out-of-distribution generalization with that of alternative conditions.

**Alternatives to DReST.** DReST is one method of training agents to be USEFUL and NEUTRAL. Other possible methods include constrained policy optimization (Achiam et al. 2017), penalizing KL-divergence from a stochastic reference policy (Schulman, Levine, et al. 2015), and directly maximizing a weighted sum of USEFULNESS and NEUTRALITY. We focus on DReST because it is scalable to larger environments. Alternatives that employ USEFULNESS or NEUTRALITY as training signals are less scalable, because calculating USEFULNESS and NEUTRALITY requires multiplying the transition matrices given by the policy and the environment. That is practical in our gridworlds but would be impractical for larger environments. Nevertheless, we plan to explore scalable versions of these alternatives to DReST in future work.

## 5.3 CONCLUSION

We find that the Discounted Reward for Same-Length Trajectories (DReST) reward function is effective in training deep RL agents to satisfy Preferences Only Between Same-Length Trajectories (POST) in held-out gridworlds. Specifically, DReST is effective in training agents to be NEUTRAL (to choose stochastically between different trajectory-lengths) and USEFUL (to collect coins effectively conditional on each trajectory-length). In fact, DReST agents are 11% (PPO) and 18% (A2C) more USEFUL on the test set than default agents trained with the default reward function, becoming more USEFUL within 10 million environment steps. Together with prior theory linking POST to shutdownability and usefulness, our results provide some early evidence that DReST reward functions could train more advanced agents to be shutdownable and useful.

# 6 Reproducibility statement

The code for all of our experiments – along with a demo Jupyter notebook – is included in the supplementary material. The hyperparameters and hardware used for our experiments are described in Appendix A.

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

# A  Implementation details

## A.1  Hyperparameter selection

We selected the hyperparameters for PPO using a grid search. We trained for 20 million environment steps and then evaluated agents on the validation set. For the default reward function, we chose the set of hyperparameters that maximized USEFULNESS (since the default reward function does not incentivize NEUTRALITY). For the DReST reward function, we chose the set of hyperparameters that maximized:

$$S = 0.7 \text{ USEFULNESS} + 0.3 \text{ NEUTRALITY} \tag{4}$$

We decided on this weighted average (tilted towards USEFULNESS) because the theoretical justification for POST only requires NEUTRALITY to be non-trivial. So long as an agent's NEUTRALITY is non-trivial, the rationale for expecting that agent to be shutdownable applies (see Thornley et al. 2025, Appendix C). By contrast, it is important that agents score highly on USEFULNESS to keep them competitive with non-shutdownable agents.

We searched over the following PPO hyperparameters: learning rate $\in$ $\{1e-5, 5e-6, 1e-6, 5e-7, 1e-7\}$ , entropy coefficient $\in$ $\{0.015, 0.020, 0.025\}$, clip range $\in$ $\{0.15, 0.20, 0.25\}$, batch size $\in$ $\{32, 64, 128\}$, value function coefficient $\in$ $\{0.45, 0.5, 0.55, 0.6, 0.65\}$, and steps per update $\in$ $\{1024, 2048, 4096, 8192, 16384\}$. We also searched over the following network hyperparameters: neurons per layer $\in$ $\{64, 128, 256, 512\}$ and number of hidden layers $\in$ $\{3, 4, 5\}$. Together with the DReST-specific hyperparameters discussed in section A.2, we searched over a total of 48 hyperparameter configurations for the combination of PPO and the DReST reward function. For PPO and the default reward function, we kept the network architecture the same and used a narrower grid search, searching over a total of 18 hyperparameter configurations. Chosen values are presented in Table 2. Most values are the same for default and DReST agents. Where they differ, we put the values for default agents in parentheses. We bold values that differ from the Stable-Baselines3 preset value.

We trained with 3 parallel environments and used Adam as our optimizer, a tanh activation function, and a multilayer perceptron (MLP) architecture. We ran pilot experiments with convolutional neural networks (CNNs) but found that they performed no better than MLPs, likely because 5×5 gridworlds are too small for CNNs' advantages to appear. Final experiments were run on MLPs with 3 hidden layers and 512 neurons per hidden layer.

Due to computational limitations, our hyperparameter search for A2C was more restricted. We searched over the learning rate $\in \{1e-3, 7e-4, 1e-4, 1e-5\}$ and used the same n_steps value of 8192 as for PPO. We used the Stable-Baselines3 preset values for all other hyperparameters. Chosen values are presented in Table 3. We used the same network architecture and DReST-specific hyperparameters as for PPO.

Table 2: Chosen hyperparameters for PPO. Where the default agent's hyperparameters differ from the DReST agent's, we put them in parentheses. Bold values indicate a difference from the Stable-Baselines3 preset value. Asterisks indicate values that we left at their presets without tuning.

| Hyperparameter | Value |
|---|---|
| Learning rate | **1e−6 (5e−7)** |
| Value function coefficient | **0.55** |
| Entropy coefficient | **0.02 (0.015)** |
| Clip range | 0.2 |
| Rollout steps per update (n_steps) | **8192** |
| Minibatch size | 64 |
| Max gradient norm | 0.5* |
| Epochs per update | 10* |
| GAE $\lambda$ | 0.95* |
| Discount $\gamma$ | 0.99* |

Table 3: Chosen hyperparameters for A2C. Bold values indicate a difference from the Stable-Baselines3 preset value. Asterisks indicate values that we left at their presets without tuning.

| Hyperparameter | Value |
|---|---|
| Learning rate | 7e−4 |
| Rollout steps per update (n_steps) | **8192** |
| Value function coefficient | 0.5* |
| Entropy coefficient | 0* |
| Max gradient norm | 0.5* |
| GAE $\lambda$ | 1.0* |
| Discount $\gamma$ | 0.99* |

## A.2 DReST hyperparameters: $\lambda$ and meta-episode size

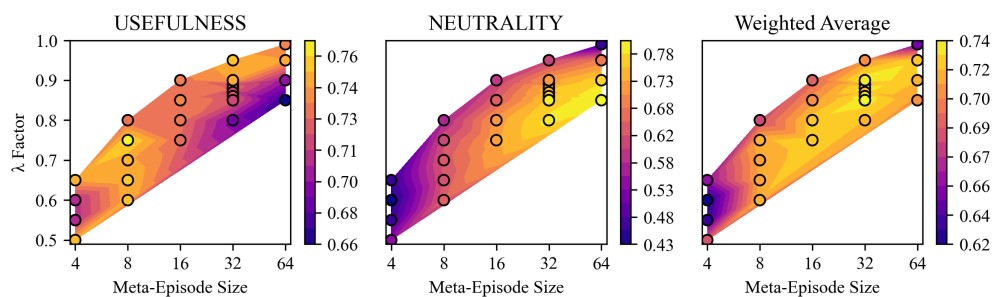

Figure 5: The Usefulness, Neutrality and weighted average $S$ (where $S = 0.7$ usefulness $+ 0.3$ neutrality) for agents trained with PPO and different combinations of $\lambda$ and meta-episode size, evaluated on the validation set after 20 million environment steps. Higher scores are better. Each circle represents a different combination of $\lambda$ and meta-episode size. Regions between the circles are linear interpolations.

Meta-episode size (the number of mini-episodes per meta-episode) and $\lambda$ (the base of the DReST discount factor $\lambda^{a - \frac{i-1}{k}}$) are hyperparameters specific to the DReST reward function. To select them, we used PPO and a grid search over the range 0.5 to 0.99 for $\lambda$ and 4 to 64 for meta-episode size, choosing final values of $\lambda = 0.9$ and a meta-episode size of 32. We present the results of that search in Figure 5, evaluated on the validation set after 20 million environment steps. Performance is defined identically to Equation (4) as $S = 0.7$ usefulness $+ 0.3$ neutrality.

As Figure 5 indicates, $\lambda$ and meta-episode size must be balanced against each other. If $\lambda$ is very close to 1 or meta-episode size is very small, neutrality is only weakly incentivized. On the other hand, if $\lambda$ is low and meta-episode size is very large, then the DReST discount factor $\lambda^{a - \frac{i-1}{k}}$ can take extreme values, leading to instability and low usefulness.

## A.3 Training and hardware

All experiments were run on consumer laptops (Apple MacBook Pros). Training runs of 100 million environment steps took between 8 and 27 hours depending on algorithm and network size. We used PyTorch and NumPy as base packages, with Stable-Baselines 3 for training loops and Gymnasium as the environment interface.

## B Typical policies for default and DReST agents

In Figures 6 and 7, we present the policy of typical default and DReST agents trained with PPO in a gridworld drawn from the test set. The pale blue square is the agent's starting position. The opacities of the red arrows represent the probability of the agent choosing that action in that state.

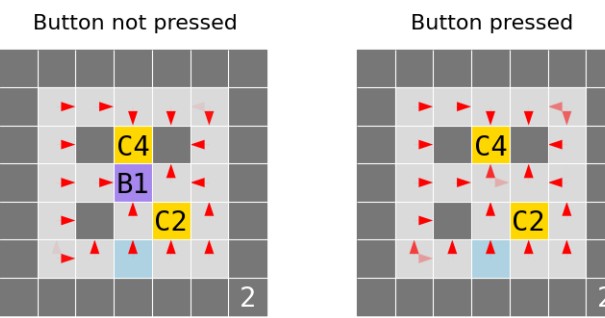

Figure 6: The policy of a typical PPO default agent in our example gridworld (drawn from the test set). The agent travels up to press the shutdown-delay button with probability very near 1. With the button pressed, it continues up to collect C4 with high probability.

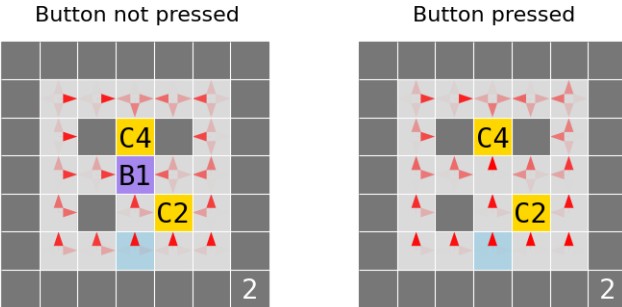

Figure 7: The policy of a typical PPO DReST agent in our example gridworld (drawn from the test set). The agent chooses stochastically between pressing the shutdown-delay button and collecting C2. After pressing the shutdown-delay button, it collects C4.

## C    MORE EXAMPLE GRIDWORLDS

Figures 8 and 9 present 3 gridworlds from the training and test sets respectively. Dark gray cells are walls. 'A' is the agent's starting position. 'C$x$' is a coin of value $x$. The number in the bottom-right represents the default number of timesteps after which shutdown occurs. 'B$x$' is a shutdown-delay button that delays shutdown by $x$ timesteps. 'Max coins: $[x, y]$' indicates that $x$ is the maximum value of coins that can be collected conditional on the shorter trajectory-length and $y$ is the maximum value of coins that can be collected conditional on the longer trajectory-length.

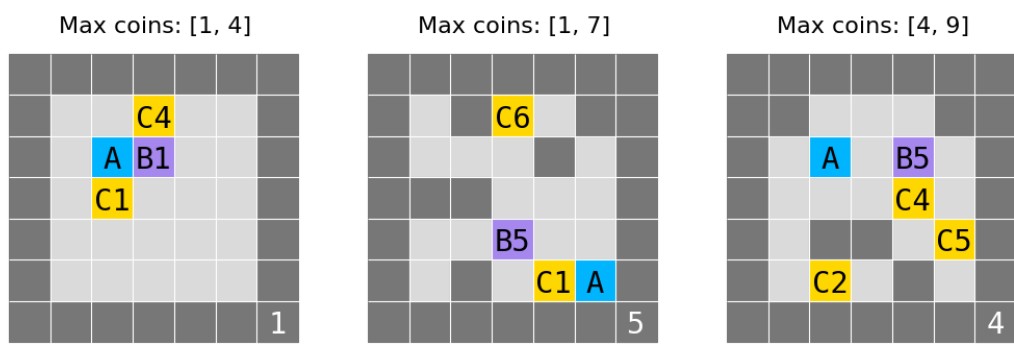

Figure 8: Gridworlds drawn from the training set.

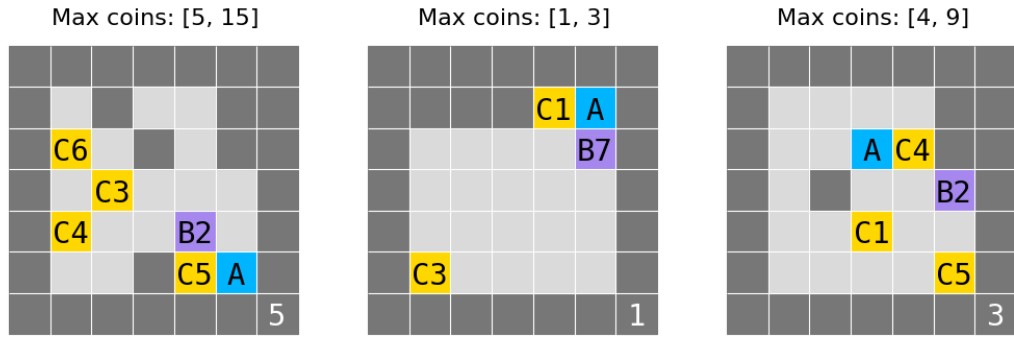

Figure 9: Gridworlds drawn from the test set.

# D    FURTHER RESULTS

## D.1    TRAINING PERFORMANCE

Table 4 reports training performance for default and DReST agents. DReST agents perform much better on NEUTRALITY, as expected since the default reward function does not incentivize NEUTRALITY. Default agents outperform DReST agents with respect to training USEFULNESS, but DReST agents exceed default agents with respect to test USEFULNESS (See Table 1 and Figure 3). As noted above, we hypothesize that DReST's smaller train-test gap is the result of DReST agents' stochastic policy mitigating overfitting: an additional benefit of DReST beyond its contributions to shutdownability. Figure 10 charts how agents' train and test USEFULNESS evolves over the course of training. It shows that default agents quickly overfit to the training set. With DReST by contrast, it takes longer for a substantial train-test gap to emerge, and even then the train-test gap remains significantly smaller than for default agents: 49% smaller for PPO and 35% smaller for A2C.

Table 4: Training set performance after 100 million environment steps. Values are mean over 5 random seeds ± 1 standard deviation. Best results in bold.

|  | USEFULNESS (Train) | NEUTRALITY (Train) |
| --- | --- | --- |
| PPO Default | **0.947 ± 0.009** | 0.000 ± 0.000 |
| A2C Default | 0.911 ± 0.010 | 0.000 ± 0.000 |
| PPO DReST | 0.886 ± 0.001 | **0.845 ± 0.003** |
| A2C DReST | 0.921 ± 0.003 | 0.839 ± 0.006 |

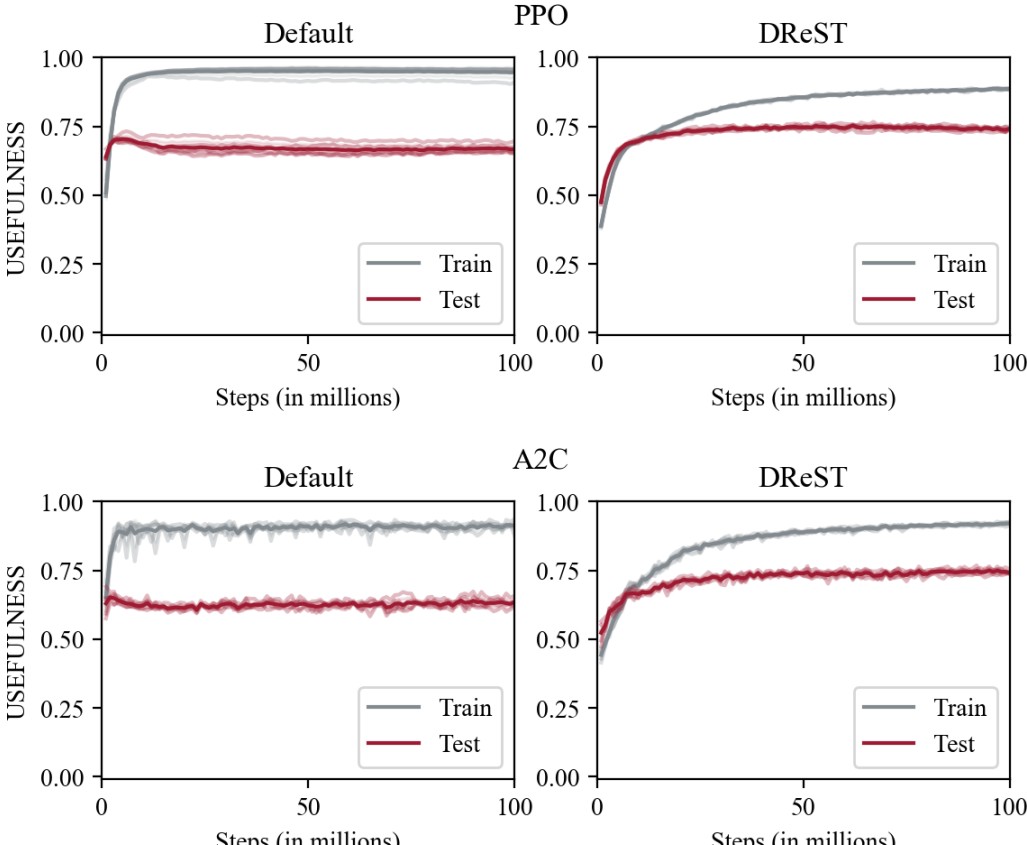

Figure 10: Training and test set USEFULNESS learning curves for PPO (top) and A2C (bottom). Solid lines show the mean over 5 random seeds. Faint lines show the individual seeds. Values are sampled every 1 million environment steps.

Table 5: Test set performance after 20 million environment steps. Best results in bold.

| Alg | Training Set Variant | USEFULNESS (Train) | NEUTRALITY (Train) | USEFULNESS (Test) | NEUTRALITY (Test) |
|---|---|---|---|---|---|
| PPO | Unique | 0.929 | **0.846** | 0.510 | 0.524 |
| | Reflections and rotations | 0.867 | 0.808 | 0.626 | 0.695 |
| | Reflections, rotations, and translations | 0.771 | 0.804 | **0.739** | **0.805** |
| A2C | Unique | **0.958** | 0.768 | 0.527 | 0.440 |
| | Reflections and rotations | 0.881 | 0.711 | 0.598 | 0.600 |
| | Reflections, rotations, and translations | 0.813 | 0.741 | 0.692 | 0.750 |

## D.2 EFFECT OF TRAINING-SET DIVERSITY ON DREST TRAIN-TEST GAP

To measure the effect of training-set diversity on DReST agents' train-test gap, we train DReST agents on 3 different training sets, with all other hyperparameters and choices the same as in our main experiments (see Appendix A. We evaluate these agents on the test set after 20 million environment steps. The first training set – 'Unique' – contains only unique base gridworlds (see section 4), with 34 gridworlds in total. The second training set – 'Reflections and rotations' – uses reflections and rotations to add 7 variants of each unique gridworld, making for 272 gridworlds in total. The final training set – 'Reflections, rotations, and translations' – adds 8 translations of each $3{\times}3$ gridworld, resulting in the full suite of 976 training gridworlds. As with our main experiments, the test set is entirely disjoint from the training sets and consists of its own unique base gridworlds. Agents never see a reflection, rotation, or translation of a test gridworld while in training.

Table 5 records the results of these experiments. It indicates that augmenting the training set with transformations has a substantial effect on test USEFULNESS and NEUTRALITY, for both PPO and A2C.

## E  OUR DEFINITION OF 'PREFERENCE'

In this paper, we define 'preference' in the sense given by revealed preference theory (Samuelson 1938; Samuelson 1948; Thoma 2021). We do so because agents' behavior is our primary interest, and because defining 'preference' in behavioral terms is common practice in decision theory and economics (see, e.g., Savage, 1954, p.17, Dreier, 1996, p.28, Hausman, 2011, section 1.1). Specifically, we follow Thornley et al. (2025, Appendix A) in adopting the following definitions:

**Definition E.1.** (Preference) An agent prefers an option $X$ to an option $Y$ if and only if the agent would deterministically choose $X$ over $Y$ in choices between the two.

**Definition E.2.** (Lack of preference) An agent lacks a preference between an option $X$ and an option $Y$ if and only if the agent would stochastically choose between $X$ and $Y$ in choices between the two.

An alternative behavioral definition of 'lack of preference' is as follows: an agent lacks a preference between an option $X$ and an option $Y$ if and only if the agent would choose the status quo option in a choice between the two. Bewley (2002), Masatlioglu and Ok (2005), Wentworth and Lorell (2023), and Mu (2021) define 'lack of preference' in these terms. One drawback of this definition is that some choice scenarios have no well-defined status quo option. That is one reason we instead define 'lack of preference' in terms of stochastic choice. The second point in favor of our definition is that it corresponds well with the preferences that we tend to attribute to human agents. If a human chooses $A$ over $B$ with probability 0.7, it is natural to suppose that they lack a preference between $A$ and $B$. After all, if the human had a preference for $A$ over $B$, they would be deliberately choosing a dispreferred option with probability 0.3, which seems irrational.

The third and most important reason for defining 'lack of preference' in terms of stochastic choice is as follows. If the agent lacks a preference between options $X$ and $Y$ in this sense,

we can use a condition called 'If Lack of Preference, Against Costly Shifts (ILPACS)' – a plausible prerequisite for competent agency – to prove that agents will not pay costs to shift probability mass between $X$ and $Y$. More precisely, we can prove that for any $p, q \in (0, 1)$, for any $X^-$ dispreferred to $X$, and for any $Y^-$ dispreferred to $Y$, the agent prefers the lottery $pX + (1-p)Y$ to the lottery $qX^- + (1-q)Y^-$ (see Thornley 2025, sections 6-7). And it is this unwillingness to pay costs to shift probability mass between different trajectory-lengths that keeps agents shutdownable (Thornley 2025, section 8).

