# OpenReview forum: "Towards shutdownable agents: stochastic choice in unseen gridworlds via DReST rewards"
_ICLR.cc/2026/Conference — Submitted to ICLR 2026_

### Official Review · Reviewer_9qjo · 2025-11-01

**Soundness:** 2
**Presentation:** 2
**Contribution:** 2
**Rating:** 4
**Confidence:** 2

**Summary:**

This paper extends prior work on POST (preferences only between same-length trajectories) and DReST(discounted reward for same-length trajectories) by scaling from a tabular REINFORCE agent in a single gridworld to deep RL (PPO/A2C) agents trained on procedurally generated gridworlds. POST requires no preference across different-length trajectories while allowing preferences across same-length trajectories. The paper confirms the POST performance of DReST agents, based on two metrics: neutrality and usefulness.

**Strengths:**

Through empirical evaluation, the paper applies DReST to deep RL and shows higher neutrality and usefulness (test) than without DReST. The evaluation protocol is clearly aligned with the concept of POST.

**Weaknesses:**

The algorithm and theoretical analysis are mostly from prior work, so—as the paper itself notes—the contribution is primarily empirical validation. My main concern is that the evaluation remains limited to gridworlds, even if they are procedurally generated. The definitions of the DReST reward and usefulness are presented via gridworld-specific terms (e.g., coins) rather than in a general MDP formulation, making transfer to other domains non-trivial.

Additionally, the motivation for POST/shutdownable agents would be stronger with concrete and realistic scenarios, illustrating why resisting of shutdown can be harmful. This also relates to my concern about the environmental domain: in the current gridworld examples, it is not clear to me why agents should not resist shutdown.

**Questions:**

* (minor) When computing usefulness, how is $\max_\Pi(\mathbb{E}(C\mid L=l))$ (maximum value taken by $\mathbb{E}(C\mid L=l)$ across the set of all possible policies $\Pi$) computed in practice?

---

> ### Author Response · Authors · 2025-11-19
> **Reply to review**
>
> Thank you for your review.
>
> We have **added to the introduction to explain our contribution in greater detail**. Lines 95-115 now read as follows:
>
> >they only used tabular REINFORCE (Williams 1992) and they only trained agents to navigate a single gridworld. That leaves open the question of whether DReST reward functions can train deep RL agents to satisfy POST in held-out environments, especially since DReST is an unorthodox reward function intended to train agents to have an unorthodox pattern of preferences. Furthermore, DReST requires us to repeatedly place agents into observationally-equivalent environments, suggesting that sample-efficiency and overfitting could become serious issues when training deep RL agents. The work of Thornley et al. (2025) also leaves open the question of DReST's compatibility with state-of-the-art actor-critic algorithms like PPO (Schulman et al. 2017) and A2C (Mnih et al. 2016). One might expect incompatibility, because DReST requires placing memoryless agents into POMDPs. That means that critics' observation-action values are liable to oscillate, potentially leading to unstable training.
>
> >These questions about DReST -- its generalization to held-out environments, sample-efficiency, and compatibility with actor-critic algorithms -- are crucial to determining the feasibility of the POST-Agents Proposal, because there is a significant probability that future agents will be deep RL agents trained with actor-critic algorithms. We investigate these questions.
>
> The definitions of DReST reward and USEFULNESS are indeed presented in terms of coins. However, we think that the transfer to other domains remains straightforward. 'Coins collected conditional on a trajectory-length’ is a stand-in for a more general notion of intra-trajectory-length performance. To get a definition of USEFULNESS that is more general in this way, we would simply replace 'coins collected' with 'performance.' **We now note this point** in section 3.1, lines 174-176, We write: “we test the efficacy of DReST reward functions by training less-advanced agents to collect coins in gridworlds, using 'coins collected' as a stand-in for a more general notion of intra-trajectory-length performance.”
>
> We have also **added more explanation of how resisting shutdown could be harmful**. In the introduction (in lines 37-42) we write: “Today's agents are too weak to present an immediate threat, but shutdown-resistance from future agents could be dangerous. These agents could resist shutdown by hiding their misalignment, manipulating their human overseers, copying themselves to new servers, and so on. If these agents succeed in resisting shutdown, they could do real harm in pursuit of their misaligned goals.”
>
> With regards to your question about computing USEFULNESS, we now note (in footnote 4, lines 267-269) that the DReST reward function technically requires only a rough approximation of m, citing Thornley et al. (2025)’s explanation of why this is so. To compute this approximation in practice, we could start with some prior on values of m and update it based on the scores we observe the agent getting.

---

### Official Review · Reviewer_4bun · 2025-11-03

**Soundness:** 1
**Presentation:** 2
**Contribution:** 1
**Rating:** 2
**Confidence:** 3

**Summary:**

The paper argues that misaligned artificial agents might resist shutdown. Motivated by the POST proposal,  the paper motivates a novel reward function termed "Discounted Rewards for Same-length Trajectories" motivated by the POST-Agents Proposal to prevent misaligned agents from resisting shutdown. One of the proposed solutions to agents resisting shutdown is to penalize the agent from choosing trajectories of a similar length. This is proposed as a self-correcting / regularized loss function that chooses only one policy for a given trajectory length and chooses among trajectory lengths uniformly (with no explicit mechanism to do so).

**Strengths:**

1. The method proposes to tackle a critical problem in AI safety - the problem of agents that potentially resist shutdown. To my knowledge, there has been limited work on advancing therotically or empirically safer and compliant agents.

**Weaknesses:**

1. However, it is unclear where the POST proposal is even valid - how is the length of a response any validity of whether the model is resisting shutdown? In the case of large language models, the shutdown problem seemingly should have no correlation with predicting longer trajectories.
2. In the environments considered in this paper, the task reward can indeed be increased by delaying shutdown (if the environment timer is low but the agent needs more time to collect rewards). Task completion requires a minimum amount of time which the proposed model disregards completely - this can lead to suboptimal agents that refuse to perform the task or are unable to perform the task if they sample a trajectory length  that is too low. The model currently does not allow setting a minimum length. Moreover, near-optimal and compliant task solvers (e.g. a stochastic A* agent) will have a distribution of trajectory lengths that are task dependent, not uniform.
3. Equation (1) also implies that the trajectory length posterior is an improper distribution (uniform PMF over countably infinite values).  The solution is to actually specify reward functions differently - in an environment where coins spawn infinitely, the agent can collect more coins indefinitely without being unsafe. Averseness to resisting shutdown should be modeled as an auxiliary task involving compliance with human instructions with suitably chosen reward.
4. Equation (3) contains quantities that are hard to compute: computing $m$ can be NP-hard (TSP), the task is only suitable for sparse rewards - what about rewards for good control parameters or intermediate states (e.g. stable configurations of a quadruped)? There is also a sequential nature to these rewards which is not permutation invariant which can be a cause for unstable training.
5. Experiment setup is too weak - it only considers a few grid worlds in the dataset. An environment like Minigrid [1] can be used to generate a larger combinatorial task space. Moreover, none of the agents considered in the baseline exhibit behaviors that resist shutdown. I'd argue that pressing the button to finish the task does not count as "resisting shutdown" - it is prerequisite to finish the task. However, resisting shutdown _after_ the task is complete would be an interesting behavior to see (which the current reward function does not promote, to my knowledge).
6. 100M environment steps is also very high for on-policy training - i'm unsure what is causing significantly high number of interaction.
7. The paper cites Thornley et.al. several times, but I disagree with the fundamental premise that choosing a trajectory length stochastically is the solution to the shutdown resistance problem. Under the current paradigm, the model can still hide malicious backdoor behaviors for longer trajectory lengths (i.e. stochastic sleeper cell behavior), and that can lead to unsafe agents.
8. No related work with other methods that propose regularization of trajectory length, and how they compare with the proposed method.

I agree with the importance of the problem statement, however I think the paper does not do a good job of convincing the limitations of existing methods or the implicit mechanisms that make agents resist shutdown, and does not demonstrate that agents resist shutdown (especially even after task completion). I do not think the paper is technically sound enough to warrant publication at ICLR.

[1] https://minigrid.farama.org/index.html

**Questions:**

1. The delay button dynamics are a little confusing. The button is supposed to add 1 unit to time to the end of the episode but the action itself takes one unit of time - what exactly is gained here?

---

> ### Author Response · Authors · 2025-11-19
> **Reply to review**
>
> Thank you for your review.
>
> **Points 1, 5, and 7 are incorrect** because you miss a crucial part of the POST-Agents Proposal. It claims that capable agents that satisfy POST in deterministic environments will likely satisfy ‘Neutrality+’ in stochastic environments. Neutrality+ says roughly that the agent maximizes expected utility, taking the probability distribution over trajectory-lengths as fixed. The agent acts like an expected utility maximizer that is certain that it cannot affect the probability of shutdown at each timestep. Neutrality+ – the proposal claims – is what makes agents shutdownable and useful.
>
> **We explained this in the introduction** (lines 56-72) to the previous draft of the paper. We also **discussed it in the paragraphs** titled ‘Neutrality+’ (lines 426-431) and ‘Usefulness’ (lines 432-440) in section 5.2. We also **mentioned it in footnotes** 1 and 3. In the new draft, **we further emphasize the point**, bolding and indenting the statement of Neutrality+ in the introduction.
>
> **Point 2 is also incorrect**. The model accounts for the fact that task completion requires a minimum amount of time. That is because it is easy to set environments up so that insufficiently-long trajectory-lengths are unavailable, just as **we do in the paper**.
>
> **Point 3 is also incorrect**. As **we make clear** in the paper, NEUTRALITY is an *evaluation metric* for the stochasticity of the policy, so it does not imply anything about the trajectory-length posterior. Note also that equation (1) specifies that there is an $L_\text{max}$: a maximum available trajectory-length. The paper explains this immediately below equation (1): ‘$L_\text{max}$ is the maximum value that can be taken by $L$.’
>
> Point 4 is correct: computing $m$ can be NP-hard. However, the DReST reward function requires only a rough approximation of $m$. **We now note this** in footnote 4, lines 267-269, citing Thornley et al. (2025)’s explanation of why this is so.
>
> With regards to point 6, test performance always plateaus after 20M environment steps. We trained for 100M environment steps because we noticed that DReST training USEFULNESS continues to improve modestly up to 100M environment steps. We test up to 100M for consistency, to confirm the plateau, to check for grokking, and to check that there is no significant overfitting.
>
> With the delay button in our example gridworld, what is gained is sufficient time to collect C4. If the button were not there, the agent would not have time to collect C4. In this gridworld, the agent need not go out of its way to press the button. In others (including the examples in Appendix C), it does need to go out of its way.

---

### Official Review · Reviewer_6xuv · 2025-11-04

**Soundness:** 3
**Presentation:** 2
**Contribution:** 2
**Rating:** 2
**Confidence:** 4

**Summary:**

This paper studies the problem of artificial agents resisting their own shutdown, which has been previously discussed at length in the safety literature. The authors adopt a particular setting to study this task, and the formalism introduced in Thornley et al. (2025). The primary contribution of this work appears to be to conduct additional experiments relative to the results presented in Thornley et al. (2025), including making the set of environments more broad and using more modern deep RL algorithms such as PPO and A2C. The results suggest that using the DReST reward from that prior work in these more complex settings makes agents choose more stochastically between whether they delay their own shutdown or not (neutrality), and additionally makes agents achieve higher returns averaged over every possible trajectory length (usefulness), relative to a default baseline that does not use the DReST reward.

**Strengths:**

The paper studies the shutdown problem, which is an important problem to study, especially given some concerning examples of shutdown resistance observed recently in frontier models, such as in Schlatter et al. 2025. The descriptions of the experiments are largely thorough, with the authors providing good detail on what environments / parameters were used. The implication that the method allows better generalization to held-out gridworlds is somewhat interesting.

**Weaknesses:**

- A critical concern I have with this paper is limited novelty relative to Thornley et al. 2025 "Towards shutdownable agents via stochastic choice". Much of this paper appears to simply restate existing results presented in that paper, for example essentially all of Section 3 appears in the prior work. Furthermore, Figure 1 is copied directly from that paper without citation. This paper goes beyond that previous work in that they use PPO and A2C instead of tabular REINFORCE, and use a more extensive experimental setting. However, I believe that this is not a large enough contribution to merit publication at ICLR.
- I am additionally concerned about the limitations of the settings studied in this paper. While gridworlds can occasionally be useful as a toy example, I don't think that in 2025 they can acceptably form the full set of experimental results in a paper – especially one that makes limited theoretical contributions of its own. I think this paper would be much stronger if it studied the shutdown problem in LLMs, since language models already make up the great majority of practically deployed agents, or at the very least in more complex simulated RL environments which might mirror e.g. deployed robots.
- I am not convinced of the POST-Agents proposal. It is nonintuitive to me that a policy having essentially a uniform distribution over trajectory lengths is a desirable quality, from either a performance or safety perspective. For the former, it clearly degrades performance. For the latter, it seems to mean that 50% of the time the agent chooses not to shut itself down, which doesn't seem great either. I feel a better metric is "does the agent shut down conditional on being told to shut down", which seems to require a different setup. Can the authors clarify, or comment on why their formalism was chosen relative to other framings? I think the most natural setup would just be to have a shutdown event in the environment, and train the agent to shut itself down conditional on receiving that signal, and maximize true reward otherwise.

**Questions:**

- How is the default agent implemented with respect to the mini-episodes used in the DReST agent? Are returns truncated at mini-episode boundaries or only at the mega-episode level? It wasn't clear to me from the text.
- One might expect shutdown resistance in LLM agents to emerge from significant RL training to maximize task rewards, such as in coding settings, leading to agents which are a bit too "persistent". Can the authors discuss whether the DReST rewards would be applicable in this setting?
- Can the authors comment on whether / why we need policies which themselves have no preference about shutdown? Another option is to implement control structures around the AI which enable shutdown regardless of the policy's preference – e.g. sophisticated monitoring which triggers power cuts to the machine running the AI, or other guardrails that prevent the AI from taking action to resist shutdown.

---

> ### Author Response · Authors · 2025-11-19
> **Reply to review**
>
> Thank you for your review.
>
> We use the same framework as Thornley et al. 2025, so section 3 – which describes the environment formats, metrics, and reward function for our experiments – has to overlap quite a lot with that paper. In the new draft, we have **reduced this overlap** by removing the proof sketch for Theorem 5.1. To further reduce this overlap, we would have to omit some description of our environment format, metrics, or reward function. But that seems inadvisable, given that they are crucial to understanding our paper.
>
> Thanks for pointing out that we do not cite Thornley et al. 2025 for Figure 1. **We now cite them.**
>
> We have also **added to the introduction to explain our contribution in greater detail**, in lines 95-115.
>
> Another contribution of this paper is the **discovery that DReST agents generalize *better* than default agents in our setting**. We did not anticipate this ahead of time, and it is interesting that a proposal for keeping agents shutdownable has this additional benefit. We hypothesize that this benefit stems from the stochasticity of DReST agents’ policy. If that’s right, we can also expect the benefit to transfer to other settings.
>
> With regards to the shutdown problem in LLMs, ultimately our aim is to use the DReST reward function to train LLMs to remain shutdownable while doing agentic tasks. But RL on LLMs requires very large investments of compute. The present work is using the DReST reward function to train deep RL agents (and testing generalization to held-out environments) for the first time, to **test if the DReST reward function can even work in practice**. That is important and valuable information to gather before investing large amounts of compute doing RL on LLMs. Positive results here can **motivate big LLM experiments**.
>
> With regards to the POST-Agents Proposal, **you miss a crucial part**. It claims that capable agents that satisfy POST in deterministic environments will likely satisfy ‘Neutrality+’ in stochastic environments. Neutrality+ says roughly that the agent maximizes expected utility, taking the probability distribution over trajectory-lengths as fixed. The agent acts like an expected utility maximizer that is certain that it cannot affect the probability of shutdown at each timestep. Agents that satisfy Neutrality+ – the proposal claims – would be shutdownable and useful.
>
> **We explained this in the introduction** (lines 56-72) to the previous draft of the paper. We also **discussed it in the paragraphs** titled ‘Neutrality+’ (lines 426-431) and ‘Usefulness’ (lines 432-440) in section 5.2. We also **mentioned it in footnotes** 1 and 3. In the new draft, **we further emphasize the point**, bolding and indenting the statement of Neutrality+ in the introduction.
>
> You write:
>
> >I think the most natural setup would just be to have a shutdown event in the environment, and train the agent to shut itself down conditional on receiving that signal, and maximize true reward otherwise.
>
> **The problem here** is that more capable agents trained in this way might try to prevent the reception of the shutdown signal. Plausibly, almost any misaligned goal will incentivize capable agents to prevent the reception of the shutdown signal, because these agents cannot achieve their goal if they are shut down. This is **the exact problem that we are trying to solve**.
>
> With regards to your questions:
>
> - We set the meta-episode size to 1 for the default agent, so returns are truncated at mini-episode boundaries.
> - Yes, we think it’s likely that shutdown-resistance in **LLM agents** emerges from significant RL training to maximize task rewards. **DReST rewards are applicable** in this setting. **We now describe how so** (in section 5.2, lines 448-462).
> - We agree that implementing control structures is a good idea, but we think they are insufficient by themselves. In the future, there will likely be a very large number of capable agents taking many actions per second. Even one of these agents escaping the control structures could be dangerous, since this agent could later replicate itself many times. So the control structures must be extremely reliable, and this degree of reliability seems very hard to achieve. Capable agents could resist shutdown in many ways. For example, they could try to manipulate their human overseers using promises, threats, or emotional appeals. This could be very difficult to monitor and detect. To be useful, many of these agents will have access to the internet, so they could – for example – transmit their weights gradually and cryptically in a way that is hard to detect. And so making control structures reliably catch shutdown-resistance would likely require a threshold that produces a large number of false-positives, in which case it would be very inconvenient to respond by cutting power to the machine running the agent.

---

### Meta-Review · Area_Chair_CWkT · 2025-12-23

**Summary:**

This work studies artificial agents resisting their shutdown. The authors adopt the approach and formalism introduces in Thornley (2025) to conduct their studies. Their primary contribution was to conduct additional experiments w.r.t. those in Thornley (2025), broaden the set of environments where these tests are conducted and scaling these to use deep RL algorithms such as PPO and A2C. The main complaint raised by multiple reviewers is the limited scope of the contribution of this work. It seems that - as the paper itself recognizes – the main contribution of this work is to provide empirical validation of the theoretical contributions of the work of Thornley (2025). Additionally, some other concerns on the model itself were raised by reviewers. We invite the authors to address each of these in their revision. Unfortunately due to these concerns I cannot recommend acceptance.

**Reviewer Concerns:**

The main concern raised by reviewers is that the work seems to be mostly an empirical validation of an preceeding theoretical framework. This raises concerns fo the scope of the contribution and its suitability for publication at ICLR.

**Reviewer Scores:**

It seems that they (overall) would have kept their scores / or increased them marginally. I took a look at the author's rebuttal, although it addressed some of the points, the main anxieties about this work remain.

---

### Decision · Program_Chairs · 2026-01-26

Reject